# Distribution Pattern and Influencing Factors for the Temperature Field of a Topographic Bias Tunnel in Seasonally Frozen Regions

**Wenbin Tang [1], Xiangdong Xu [2], Tao Zhang [3,*], Hong Wang [2,4] and Jianxing Liao [4]**

[1] China Railway Development Investment Group Co., Ltd., Kunming 650200, China; tangwenbin29@163.com
[2] Guizhou Transportation Planning Survey & Design Academe Co., Ltd., Guiyang 550081, China; xuxiangdong2023@163.com (X.X.); hwang23@gzu.edu.cn (H.W.)
[3] Yellow River Engineering Consulting Co., Ltd., Zhengzhou 450003, China
[4] College of Civil Engineering, Guizhou University, Guiyang 550025, China; jxliao@gzu.edu.cn
[*] Correspondence: zhangtao18@mails.jlu.edu.cn

**Abstract:** In seasonally frozen regions, highway tunnels are prone to shallow buried bias pressures near the inlet/outlet, which leads to highway tunnels not only bearing asymmetric loads, but also facing the threat of extreme weather. However, there is still no clear understanding of the temperature field for topographically biased tunnel in seasonally frozen regions at present. Taking the Huitougou tunnel of Hegang-Dalian expressway as the object, this paper uses on-site monitoring, theoretical analysis, and numerical simulation to study the distribution law and influence factors of temperature field for topographically biased tunnel in seasonally frozen regions. The numerical results of the temperature field are in good agreement with the on-site monitoring data, which verified the accuracy of this numerical model based on the aerodynamic principle, turbulence model, and wall function method. Meanwhile, the effect of different slope angle and overburden thickness on the temperature field of the tunnel is further analyzed. It is found that when the slope angle increases, the temperature field in the tunnel surrounding rock changes accordingly. The connecting area between the surface and the tunnel temperature field is deflected from arch crown to the arch shoulder of the tunnel, resulting in a large change in the temperature of the shallow buried side, while minor change in the temperature of the deep buried side. The freezing depth of surrounding rock decreases with the rising slope angle. As the overburden thickness gradually increases, the temperature field of the surface surrounding rock and the tunnel surrounding rock gradually change from mutual influence to non-influence. When the overburden thickness exceeds 15 m, a "isolated temperature zone" appears in the middle with a temperature of 6~7 °C, the temperature field of the tunnel surrounding rock is basically not affected by the surface air temperature. These results can provide important theoretical and engineering guidance for the evaluation, construction, and maintenance of tunnel engineering in seasonally frozen regions.

**Keywords:** seasonally frozen region; topographically biased tunnel; temperature field; numerical simulation

## 1. Introduction

In China, approximately 53% of the land area comprises seasonally frozen soil, primarily located in high altitude mountainous regions within the northeast, northwest, north, and southwest [1–3]. The construction and operation of tunnels in seasonally frozen areas are heavily influenced by frost heaving at the inlet/outlet of the tunnel, and the frequency and degree of frost damage will significantly increase with the freezing depth [4]. When a tunnel in a seasonally frozen region encounters a shallow-buried and unsymmetrical pressure section, asymmetric loads due to ground surface inclination threaten the stability of surrounding rock and support lining, and exacerbate frost damage at the inlet/outlet.

Consequently, the degree of bias pressure and the distribution characteristics of the temperature field should be considered carefully in the highway bias tunnels in seasonally frozen regions.

In comparison to general tunnel engineering, tunneling in cold regions face much more complex technical issues, with considering the impact of frost damage, requiring high anti-freezing capacity of the rock-soil mass and specifically supporting structures. To address these problems, accurately depicting the distribution, evolution, and key control factors for the temperature field surrounding tunnels is very important. Lot of research have been conducted to solve the temperature field. For instance, Lai et al. [5] proposed a thermal-hydro-mechanical (THM) coupled mathematical and mechanical model based on the Galerkin method, and found that the frost heave force has a significant impact on the stress of tunnel lining, and an approximate analytical solution for the freezing process in a circular tunnel is also given [6]. Huang et al. [7] preliminarily established a thermal-hydro (TH) coupled mathematical model with equivalent permeability and thermos-physical properties adopted in fissure surrounding rock mass, successfully applied in the Qinling Tunnel project. He et al. [8] established a comprehensive model for convective heat transfer and solid heat conduction between air and surrounding rock in the tunnel based on the meteorological conditions, and analyzed the temperature inside the tunnel, as well as the freezing and melting conditions of surrounding rock in the Dabanshan tunnel from the Qilian Mountains. Wang et al. [9] released a program using Neumann Stochastic finite element method to analyze the temperature field of tunnels in cold regions. Xu et al. [10] analyzed longitudinal temperature monitoring data of a tunnel, finding a V-shaped distribution after the number of freeze-thaw cycles. Liu [11] conducted a numerical simulation of the two-dimensional unsteady temperature field of tunnels in cold regions using the finite element method, obtaining the maximum freezing depth of surrounding rock and the relationship curve between surrounding rock temperature and time. Yang et al. [12] studied the effects of various factors on frost heave, and the results show that the vertical component of frost heave is normally distributed, with the maximum at the tunnel axis, while the horizontal component is maximum at a distance from the tunnel axis. Tan et al. [13] conducted a study by using numerical analysis methods to examine the variation law of the surrounding rock temperature field in the Galongla tunnel from Tibet under ventilation conditions, revealing that before the excavation of the tunnel, there was a considerable shift in the temperature of the shallow mountain part with seasonal changes. Additionally, the critical depth of the temperature-sensitive zone was noted to be 18 m.

Recently, many scholars have conducted research on the temperature field of tunnels in cold regions, identified the characteristics and laws of temperature distribution on the longitudinal section of the tunnel, verified the frost heave of tunnel surrounding rock, and proposed various targeted measures to prevent and control frost damage [14–16]. However, previous research still has some shortcomings as the following:

1.  For the temperature distribution on the tunnel cross section, the influence of terrain change is seldom considered;
2.  For shallow buried unsymmetrical pressure tunnels, the overburden thickness is an important factor affecting the temperature field of the tunnel surrounding rock, and current understanding is still insufficient.

To address these issues, this article takes the HTG tunnel as an engineering case, by using an integrated method combing the field investigation, on-site monitoring, and numerical analysis to study the spatiotemporal evolution law of the temperature field in surrounding rock of topographic bias tunnel and its potential relationship with terrain changes. The research results can provide a theoretical basis for the design of antifreeze measures for topographically biased tunnels in seasonally frozen areas.

## 2. Engineering Background and Methods

### 2.1. Engineering Background

The HTG Tunnel is an important part of the Hegang-Dalian Expressway, located in the border area between Tonghua County and Baishan City in Jilin Province. As shown in Figure 1, the tunnel site is located 15.7 km from Baishan City, and situated in a typical seasonally frozen area in northeast China. The left line of the tunnel is 720 m long (starting and ending pile numbers: LK315 + 655~LK316 + 375), and the right line is 660 m long (starting and ending pile numbers: RK315 + 680~RK316 + 340), belonging to a medium-sized tunnel. The longitudinal section of the entrance on right line of the HTG tunnel is a typical shallow buried unsymmetrical pressure tunnel with a buried depth of 14.49 m and a slope angle of about 21° (Figure 2a,b).

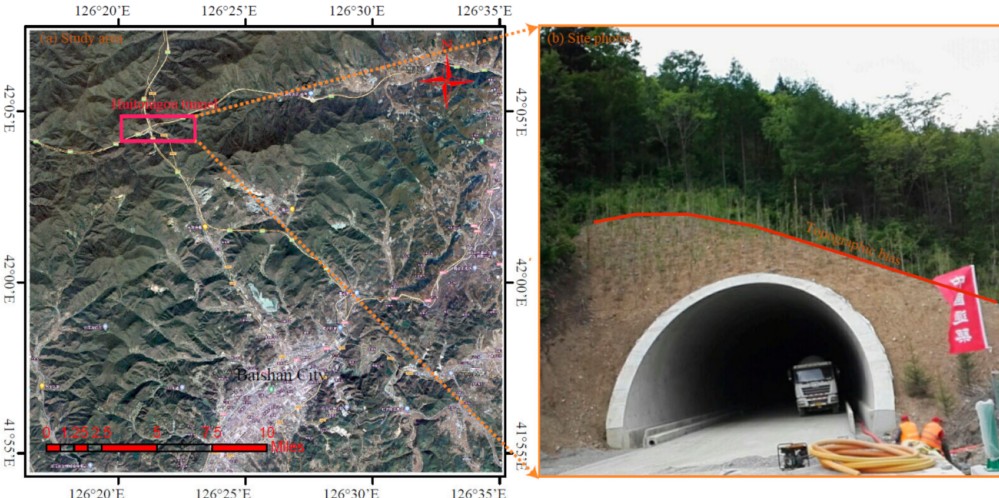

**Figure 1.** Huitougou tunnel and its temperature monitoring scheme.

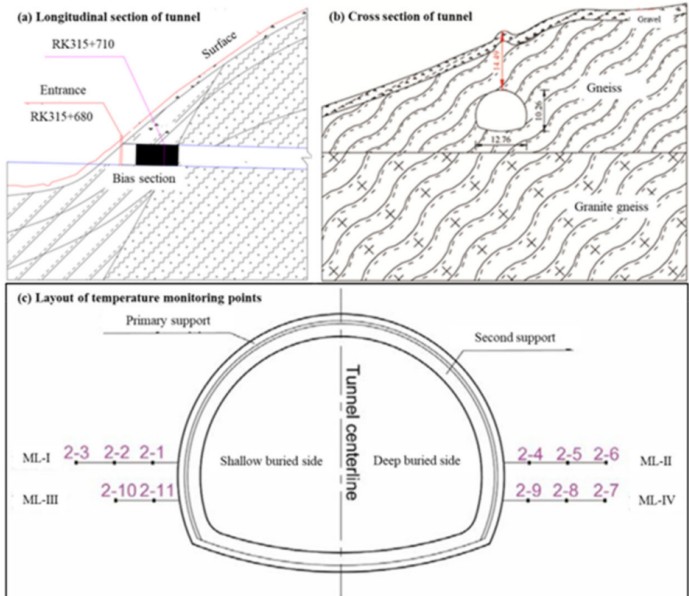

**Figure 2.** HTG Tunnel longitudinal/cross section and temperature monitoring scheme.

To monitor the temperature distribution of the terrain bias section of the right line, temperature sensors are installed on the tunnel cross section (Figure 2c). The monitoring was conducted in the period from May 2015 to June 2016. Detailed monitoring data analysis can be found in Section 3.1.

### 2.2. Numerical Analysis Method

In this paper, COMSOL Multiphysics is used to analyze the distribution characteristics of temperature field and influence factors in the shallow buried bias section of the HTG tunnel. The governing equations are given in as follows:

#### 2.2.1. Mass Conservation Equation

In the flow field, the flow of gas in the tunnel follows the law of mass conservation, and the expression is is as follows [17]:

$$\frac{\partial \rho}{\partial t} + \nabla \cdot (\rho u) = 0 \tag{1}$$

where $u$ is the wind speed in the tunnel, $\rho$ is density of air, $t$ is time.

Let the velocity components of $u$ in the $x$, $y$, and $z$ directions be $m$, $v$, and $w$. The mass conservation equation can be written as:

$$\frac{\partial \rho}{\partial t} + \frac{\partial(\rho m)}{\partial x} + \frac{\partial(\rho v)}{\partial y} + \frac{\partial(\rho w)}{\partial z} = 0 \tag{2}$$

Since density is usually constant, the above equation can be simplified as:

$$\frac{\partial(m)}{\partial x} + \frac{\partial(v)}{\partial y} + \frac{\partial(w)}{\partial z} = 0 \tag{3}$$

#### 2.2.2. Momentum Conservation Equation

For a given fluid system, the rate of change in momentum in time is equal to the sum of external forces acting on it [18]. The mathematical expression is:

$$\rho \frac{\partial u}{\partial t} + \rho(u \cdot \nabla)u = \nabla \cdot [-p \cdot I + \tau] + F \tag{4}$$

where $p$ is the pressure, $I$ is the identity matrix, $\tau$ is the shear stress, and $F$ is the volumetric force. On the three-dimensional scale, its mathematical expression is transformed into:

$$\begin{cases} \frac{\partial(\rho u)}{\partial t} + u \cdot \frac{\partial(\rho u)}{\partial x} = -\frac{\partial p}{\partial x} + \frac{\partial \tau_{xx}}{\partial x} + \frac{\partial \tau_{yx}}{\partial y} + \frac{\partial \tau_{zx}}{\partial z} + F_x \\ \frac{\partial(\rho v)}{\partial t} + u \cdot \frac{\partial(\rho v)}{\partial y} = -\frac{\partial p}{\partial y} + \frac{\partial \tau_{xy}}{\partial x} + \frac{\partial \tau_{yy}}{\partial y} + \frac{\partial \tau_{zy}}{\partial z} + F_y \\ \frac{\partial(\rho w)}{\partial t} + u \cdot \frac{\partial(\rho w)}{\partial z} = -\frac{\partial p}{\partial z} + \frac{\partial \tau_{xz}}{\partial x} + \frac{\partial \tau_{yz}}{\partial y} + \frac{\partial \tau_{zz}}{\partial z} + F_z \end{cases} \tag{5}$$

#### 2.2.3. Energy Conservation Equation

The mathematical expression for the energy conservation equation is [19]:

$$\rho C_p \frac{\partial T}{\partial t} + \nabla \cdot (-k\nabla T) = Q - \rho C_p u \cdot \nabla T + \tau : S + \frac{T}{\rho} \left( \frac{\partial \rho}{\partial T} \right)_p \left( \frac{\partial p}{\partial t} + u \cdot \nabla p \right) \tag{6}$$

where $C_p$ is the constant pressure heat capacity, and $Q$ is the heat source, $k$ is the thermal conductivity.

#### 2.2.4. Turbulence Model

It is considered that the movement of wind in a tunnel obeys the turbulence criterion and follows the $k - \varepsilon$ two equation model. The combining concept of boussinesq vorticity viscosity coefficient, turbulent kinetic energy $k$, and turbulent dissipation rate $\varepsilon$ is used to represent the eddy viscosity coefficient, the wall function is used to solve the flow field near the tunnel wall [20,21].

When using $k - \varepsilon$ two equation model, the $\mu_T$ is calculated by:

$$\mu_T = \rho C_\mu \frac{k^2}{\varepsilon} \tag{7}$$

The turbulent kinetic energy $k$ transport equation is:

$$\rho \frac{\partial k}{\partial t} + \rho u \cdot \nabla k = \nabla \left( \left( \mu + \frac{\mu_T}{\sigma_k} \right) \nabla k \right) + P_k - \rho \varepsilon \tag{8}$$

$$P_k = \mu_T \left( \nabla u : \left( \nabla u + (\nabla u)^T \right) \right) - \frac{2}{3} (\nabla \cdot u)^2 - \frac{2}{3} \rho k \nabla \cdot u \tag{9}$$

The turbulent dissipation rate $\varepsilon$ transport equation is:

$$\rho \frac{\partial \varepsilon}{\partial t} + \rho u \cdot \nabla \varepsilon = \nabla \left( \left( \mu + \frac{\mu_T}{\sigma_\varepsilon} \right) \nabla \varepsilon \right) + C_{\varepsilon 1} \frac{\varepsilon}{k} P_k - C_{\varepsilon 2} \rho \frac{\varepsilon^2}{k} \tag{10}$$

where $\rho$ is density of air; $u$ represents the wind speed in the tunnel; $t$ means time; $\sigma_k$, $\sigma_\varepsilon$, $C_{\varepsilon 1}$, $C_{\varepsilon 2}$, and $C_\mu$ are turbulence constant. For an ideal gas, the usual value is: $\sigma_k = 1.00$, $\sigma_\varepsilon = 1.30$, $C_{\varepsilon 1} = 1.44$, $C_{\varepsilon 2} = 1.92$, $C_\mu = 0.09$.

### 2.2.5. Heat Conduction

Heat conduction refers to the phenomenon of physical transfer caused by the thermal motion of microscopic particles inside an object. In one-dimensional steady-state heat conduction, heat conduction can be described by Fourier's law [22]:

$$q_x = -k \frac{dT}{dx} \tag{11}$$

where $q_x$ is the heat flux density, which represents the heat transfer rate in the $x$ direction; $T$ is the temperature.

### 2.2.6. Thermal Convection

Convection includes thermal convection and convective heat transfer, where thermal convection occurs mainly within the air and is not considered in this paper. The basic law of convective heat transfer is expressed as follows:
When the fluid is cooled:

$$Q = hA \left( T_f - T_w \right) \tag{12}$$

When the fluid is heated:

$$Q = hA \left( T_w - T_f \right) \tag{13}$$

where $Q$ is the heat flow rate; $T_f$ is the wall temperature; $T_w$ is the fluid temperature; $A$ is the surface area of convective heat transfer on the solid wall; h is the convective heat transfer coefficient.

The wall function is a commonly used method for solving convective heat transfer [23], turbulence is a non-isothermal flow, and the convective heat transfer coefficient $h$ is:

$$h = \frac{\rho c_p C_\mu^{1/4} k^{1/2}}{T^+} \tag{14}$$

where $c_p$ is heat capacity at constant pressure, $T^+$ is a dimensionless temperature, $T^+ = T^+(y^+)$. $y^+$ is the distance between the fluid domain and the wall surface, and the specific relationship between the two is as follows:

$$T^+ = \begin{cases} P_r y^+ & y^+ < y_1^+ \\ 15 P_r^{2/3} - \frac{500}{(y^+)^2} & y_1^+ \le y^+ < y_2^+ \\ \frac{\ln(y^+) P_r}{k} + \beta & y^+ \ge y_2^+ \end{cases} \tag{15}$$

where:

$$\begin{cases} y_1^+ = \frac{10}{P_r^{1/3}} \\ y_2^+ = 10\sqrt{10\frac{K}{Pr_T}} \\ \beta = 15 Pr^{2/3} - \frac{Pr_T}{2K}\left[1 + \ln\left(1000 - \frac{K}{Pr_T}\right)\right] \end{cases} \tag{16}$$

Tan et al. [24,25] demonstrated the correctness of the wall function method by comparing it with empirical formulas for calculating heat transfer between airflow and solids. The results show that the wall function method is in good agreement with the empirical formula method, especially when the inlet air velocity is less than 10 m/s.

2.2.7. Model Parameters and Boundary Conditions

According to the field investigation results and indoor tests, the model physical and mechanical parameters are shown in Table 1.

**Table 1.** Model parameters [26].

| Materials | Density/kg·m$^{-3}$ | Thermal Conductivity W/(m·K) | Heat Capacity J(m·K) |
|---|---|---|---|
| Surrounding rock | 1900 | 2.9 | 850 |
| Air | 1.29 | 0.025 | 230 |
| Lining | 2500 | 1.8 | 950 |

The monthly average temperature change from 2015 to 2016 is shown in Figure 3, and the fitting equation is Equation (17). In the course of the numerical analysis, the temperatures at the entrance and exit of the tunnel can be considered to be essentially consistent. The expression is Equation (17).

$$T_{\text{inlet}} = T_{\text{outlet}} = T = 5.4 + 17.8\left[\frac{2\pi}{365}(t - 0.02)\right] \tag{17}$$

where $T_{\text{inlet}}$ and $T_{\text{outlet}}$ represent the temperature of tunnel inlet and outlet, $t$ is time (Unit: month). When $t = 0$, it is May.

Generally, the temperature gradient of the atmosphere in the vertical direction is 0.6 °C for every 100 m increase in height. Therefore, the temperature boundary conditions above the mountain are:

$$T_{\text{surface}} = T - 0.6 * (H - H_{\text{inlet}})/100 \tag{18}$$

where $H$ is the elevation of the ground surface, $H_{\text{inlet}}$ the inlet elevation of HTG tunnel.

The initial wind speed is also an important factor affecting the temperature field of surrounding rock in tunnel. The wind speeds observed within the study do not present a distinct mathematical pattern and are highly discrete. Therefore, the mean value of the wind speed monitoring data is chosen as the initial velocity at the tunnel entrance in this paper. That is, $u_{in}$ = 5 m/s. Meanwhile, a pressure boundary is set at the tunnel exit with a pressure of 1 atm. That is, $p_{out}$ = 1 atm.

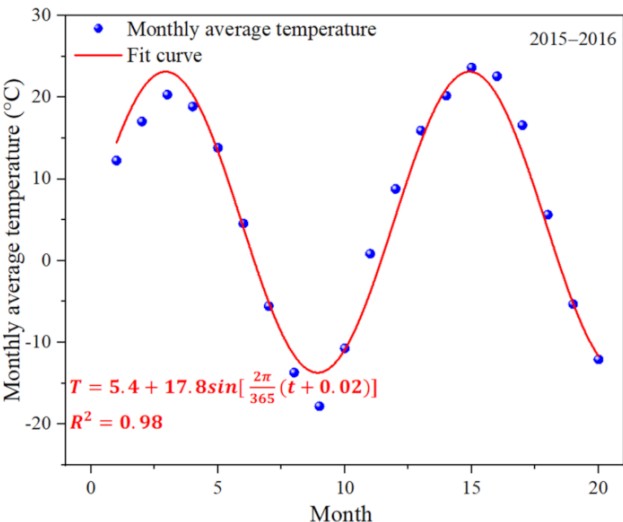

**Figure 3.** Monthly average temperature.

Since the computational area is wide enough, the boundary below the tunnel inlet and outlet is set as a thermal insulating boundary. The low edge boundary is an internal heat flux boundary with a heat flux density of 0.06 W/m². According to local data, the geothermal gradient is taken as 3 °C/100 m.

Based on the above analysis, a three-dimensional numerical model is established based on the cross section RK315+710 of HTG tunnel to analyze the variation rules and influencing factors of the temperature field for the HTG tunnel (Figure 4). The influencing factors include slope angle and overburden thickness. The mesh size is an important parameter for the accuracy of numerical simulation calculations. Referring to previous literature, this paper sets the grid size of the air flow tunnel boundary to 0.02 m~0.1 m and the rock mass grid to 0.4 m~5 m [27]. The unit of *t* in the temperature function in this paper is days, but the default unit of time in the simulation software is seconds, so we have used a time step of 86,400 s, which is one day.

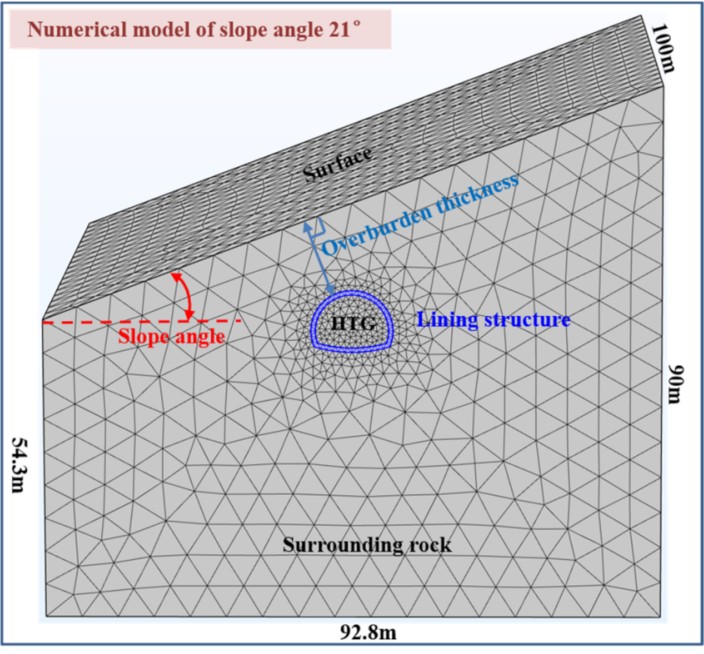

**Figure 4.** Numerical model of HTG tunnel cross section RK315+710.

## 3. Results and Discussions

### 3.1. Monitoring Analysis

The data in Figure 5 show the temperature spatio-temporal law during a monitoring period from 2015-5 to 2016-6. The temperature of the HTG tunnel surrounding rock fluctuates with the air temperature, approximately following a sinusoidal trend, with obvious freezing and thawing periods. The months with temperatures less than 0 °C are mainly January through April, which is the freezing period, and the months with temperatures more than 0 °C are mainly May through December, which is the melting period.

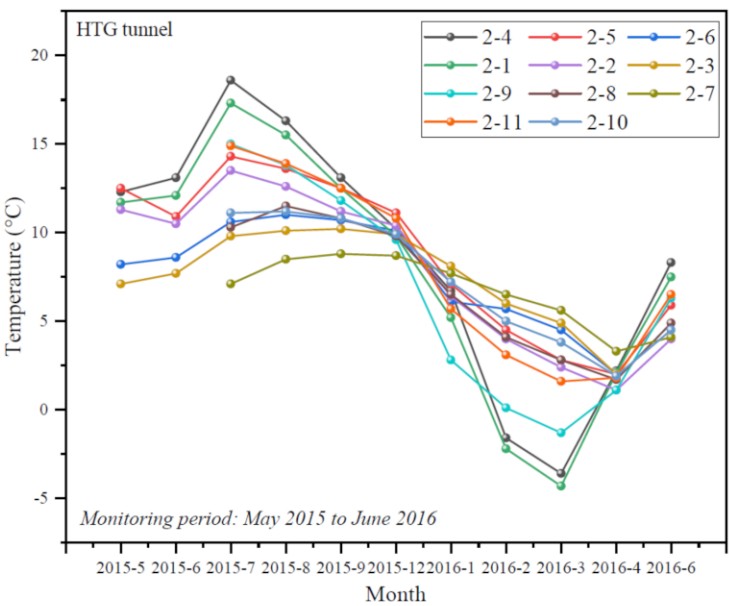

**Figure 5.** The evolution law of temperature over time.

In the melting period, as the radial distance increases, the temperature of the tunnel surrounding rock gradually decreases (Figure 6). On the contrary, in the freezing period, as the radial distance increases, the temperature of the tunnel surrounding rock gradually increases. This is consistent with the actual situation, in which the air temperature inside the tunnel is high during the melting period, while the temperature of the surrounding rock is relatively low, and the temperature propagates radially toward the surrounding rock. In the freezing period, the air temperature inside the tunnel is low and the temperature of surrounding rock is relatively high, so the temperature increases with the rising radial distance due to the effect of temperature difference.

### 3.2. Numerical Simulation

#### 3.2.1. Real Working Condition Simulation of HTG Tunnel

Taking the section in the middle of the model as the analysis object, the contour maps of the temperature for different month are shown in Figure 7. The temperature of the surrounding rock is clearly affected by the air temperature of inside the tunnel and at the surface, which varies seasonally. In January, the air temperature in the tunnel is lower than −10 °C, and the temperature of the surrounding rock increases with the distance from the tunnel wall, reaching 6 °C at about 8 m. In April, the effect of the cold air on the temperature of the surrounding rocks is continuously extended by the continuous convection of cold air within the cave. The temperature field of the surrounding rock equilibrates with the surface temperature field and the isotherm lines form a closed loop connection. With the exchange of heat over time, the distance between the 6 °C isotherm line and the tunnel wall reached 14 m. Similarly, in July, the air temperature rose to 24 °C, but the temperature of the surrounding rock has not changed. In October, the air temperature dropped and the temperature field in the surrounding rock of tunnel underwent significant changes.

The continuous convective heat transfer between the air temperature in the tunnel and the lining wall increases the temperature of the surrounding rock behind the lining and causes the surrounding rock isotherm to connect to the surface isotherm.

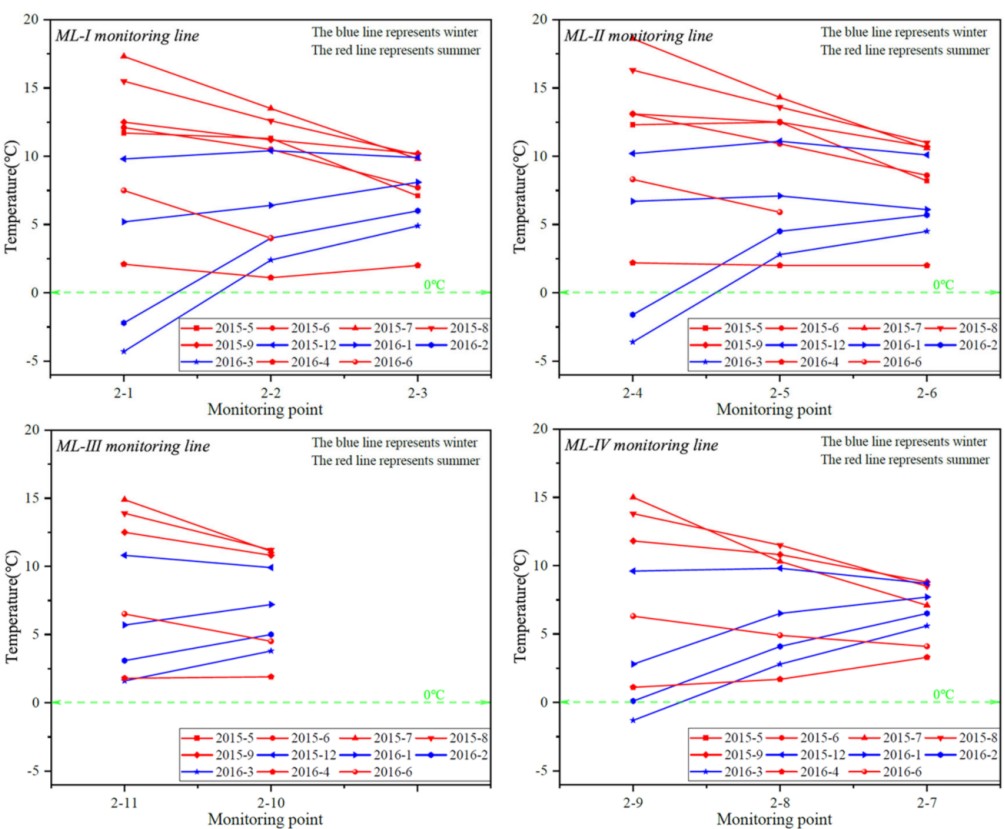

**Figure 6.** The variation law of temperature with radial distance.

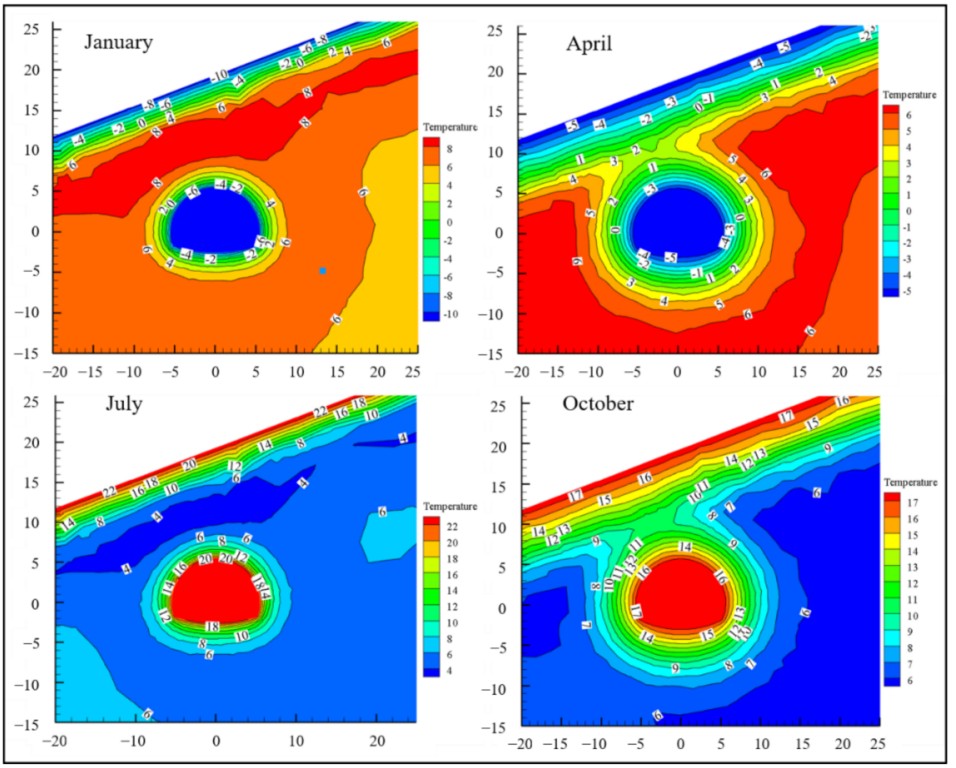

**Figure 7.** Distribution of temperature field of tunnel surrounding rock in different month.

In April and October, the tunnel surrounding rock isotherm is connected with the surface isotherm, and the temperature of surrounding rock at the junction is not equal to the initial temperature, which means that the influence area of the air temperature in the tunnel on the surrounding rock temperature field coincides with the influence area of the surface air temperature, and the influence area produces heat conduction in two directions at the same time. These results indicate that the temperature field of terrain-biased tunnels is indeed affected by changes in air temperature.

According to the actual monitoring point of the HTG tunnel, a monitoring point that is consistent with the actual monitoring set in the numerical model, and the temperature on the monitoring point of numerical model are recorded and compared with the actual monitoring results, as shown in Figure 8. It can be seen that, similar to the actual monitoring, the simulated temperature of surrounding rock in the tunnel shows a sinusoidal function trend with time, which is influenced by the atmospheric temperature (Supplementary Materials Table S1). The linear correlation index $R^2$ between the numerical analysis results and the monitoring results is 0.864, indicating that the numerical model has high accuracy and can provide a basis for subsequent analysis.

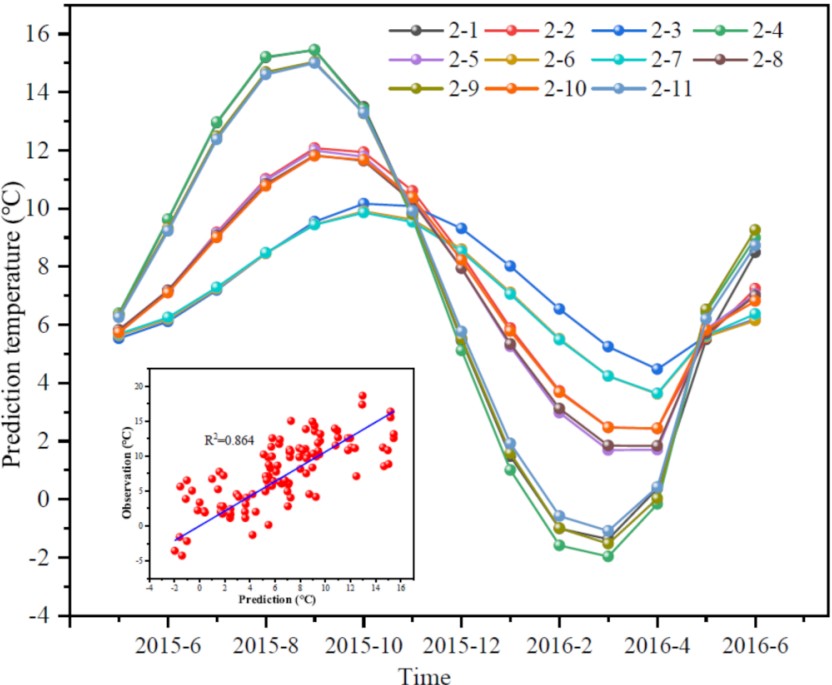

**Figure 8.** Temperature evolution law based on numerical analysis results.

Using 0 °C as the standard, determine the maximum freezing depth of tunnel surrounding rock. As shown in Table 2, the maximum freezing depth of the HTG tunnel ranges from 1.86 to 2.46, and freezing depth on shallow side slightly higher than that on deep buried side, which could be related to the influence of air temperature. In addition, the freezing depth based on the numerical analysis results is higher than the actual monitoring results, which is relatively conservative in the design and construction process.

**Table 2.** Comparison of maximum freezing depth monitoring data and simulation results.

| Number of Monitoring Line | Annual Maximum Freezing Depth/m | |
| :---: | :---: | :---: |
| | Field Monitoring Data | Simulation Results |
| ML-I | 2.13 | 2.44 |
| ML-II | 1.90 | 2.20 |
| ML-III | 2.46 | 2.37 |
| ML-IV | 2.16 | 2.34 |

### 3.2.2. Influence of Slope Angle on Temperature Field

Based on the numerical model established above, the variation pattern of surrounding rock temperature field in the melting and the freezing period under the influence of different slope angle and overburden thickness has been analyzed. The slope angle is set as 0°, 10°, 21°, 30°, and 40° respectively, and the overburden thickness is set as 5 m, 11.6 m, 15 m, and 20 m.

Figures 9 and 10 show the nephogram of the temperature field of surrounding rock under the influence of different slope angles in the melting and freezing period. When the slope angle is 0°, air temperature can affect the surrounding rock temperature, forming a connection area between the temperature of surrounding rock and the surface temperature. The isotherm in the connection area is 10 °C in the melting period and 3 °C in the freezing period. On both sides of the connection area, the isotherms are connected in a curve. The isotherms near the tunnel area are roughly parallel to the tunnel wall, while the isotherms near the surface are approximately parallel to the surface. From the tunnel wall to the surface, the temperature first decreases and then increases in the melting period, while it shows the opposite trend in the freezing period. The temperature fields are symmetrically distributed along the tunnel axis.

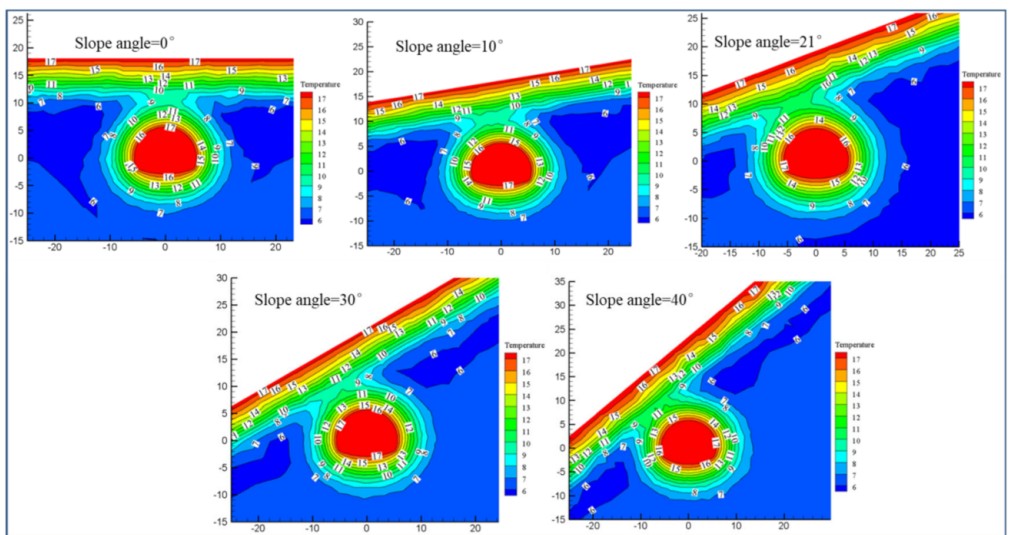

**Figure 9.** Temperature field of tunnel surrounding rock with different terrain angles in melting period.

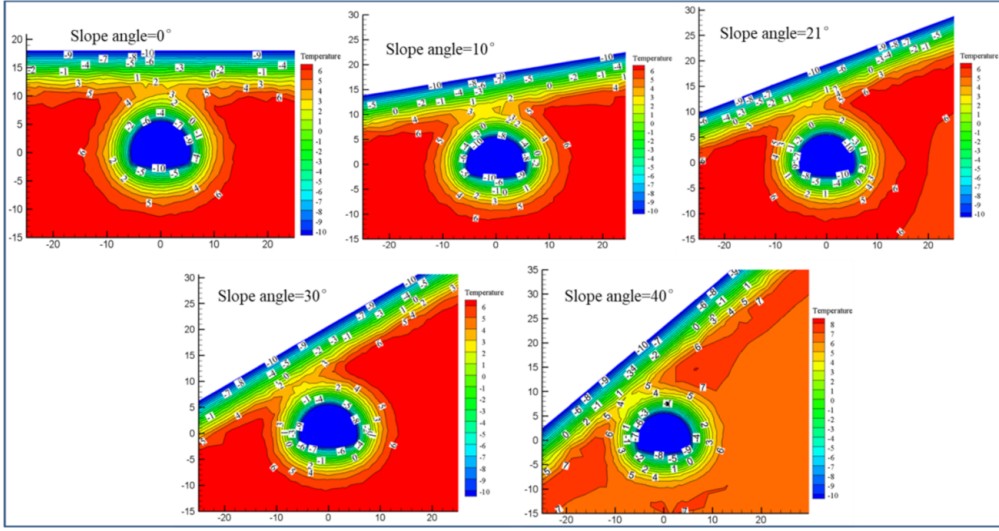

**Figure 10.** Temperature field of tunnel surrounding rock with different terrain angles in freezing period.

When the slope angle is 10°, the connection area originally located at the arch crown deflects toward the arch shoulder direction with the terrain, that is, toward the shallow buried side. The temperature in the center of connection area is 10 °C in the melting period and 3 °C in the freezing period. When the slope angles are 21°, 30°, and 40°, the connection area continuously deflects toward the arch shoulder as the slope angle increases, eventually approaches the arch shoulder, and the temperature field on both sides of the tunnel axis changes from symmetric to asymmetric distribution. However, the temperature value in the center of the connection area remains unchanged, at 10 °C in the melting period and 3–4 °C in the freezing period.

To further study the impact of the changes in slope angle on the tunnel temperature field, the temperature distributions on the monitoring lines were measured. As shown in Figure 11, the temperature of the surrounding rock continuously decreases with the distance from the tunnel wall in the melting period, and eventually tends to stabilize with distance over 8 m. When the slope angle is 40°, the temperature value of surrounding rock is significantly higher than that under other conditions. Moreover, the temperature curve corresponding to 40° experienced a rebound phenomenon on monitoring line I and III. In the freezing period, the temperature of surrounding rock increases continuously with distance from the tunnel wall, and becomes different for different slope angles when the distance exceeds 4 m. The temperature on the shallow buried side increases with the rising of slope angle, and the same trend exists on the deep buried side, but the trend is not significant enough.

Overall, the surrounding rock of tunnel is affected by the hot air inside the tunnel during the melting period, and the temperature of the surrounding rock increases due to heat absorption. The temperature of the surrounding rock decreases with distance from the tunnel wall. When the distance exceeds 8 m, the temperature of surrounding rock tends to stabilize. In the freezing period, the surrounding rock of tunnel is affected by the cold air inside tunnel, causing temperature drop of heat emitted by the surrounding rock. The temperature of the surrounding rock increases with distance from tunnel wall. At distances larger than 7 m, the temperature of the surrounding rock tends to stabilize. Within the range of 4 m from the tunnel wall, the temperature of surrounding rock is not affected by changes in slope angle. Within a distance of 4 m from the tunnel wall, the surrounding rock temperature is unaffected by the change in slope angle. For distances greater than 4 m, the larger the slope angle, the greater the temperature of the surrounding rock at the same distance. The temperature of the shallow buried surrounding rock is easily affected by the surface air temperature.

Based on the distribution characteristics of temperature field in the freezing period, the freezing depth of the surrounding rock is determined using 0 °C as the standard. As shown in Figure 12, the freezing depth on the shallow buried side is greatly affected by the slope angle, while the freezing depth on the deep buried side is slightly affected. On the shallow buried side, the freezing depth gradually decreases with rising slope angle, especially in the case with slope angle of 40°. When the slope angle of the terrain is large, the surface temperature and the temperature inside the tunnel will jointly affect the temperature field of the surrounding rock, reducing the temperature difference in the connection area, and thus reducing the freezing depth.

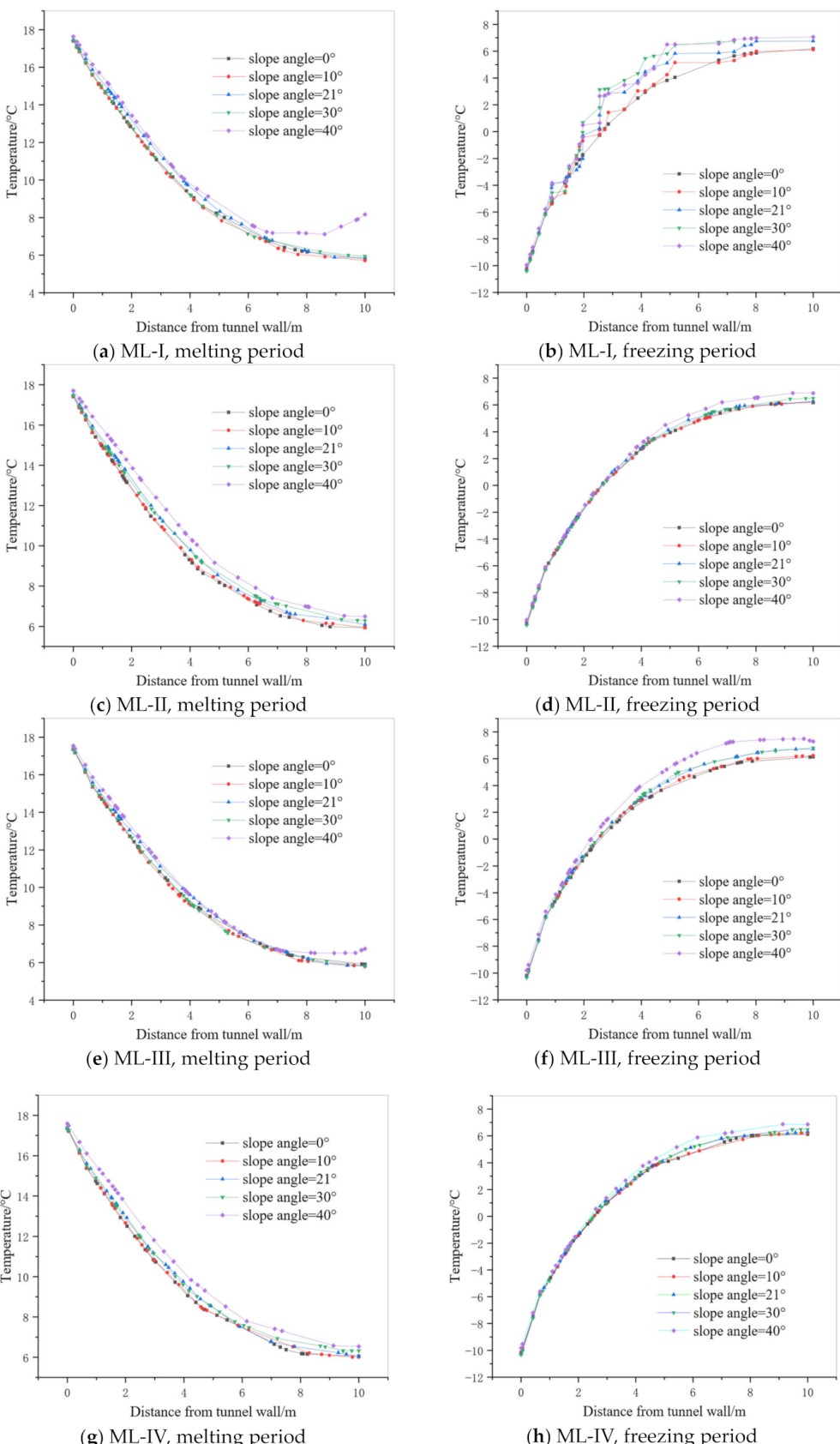

**Figure 11.** Temperature of different monitoring lines in melting and freezing period.

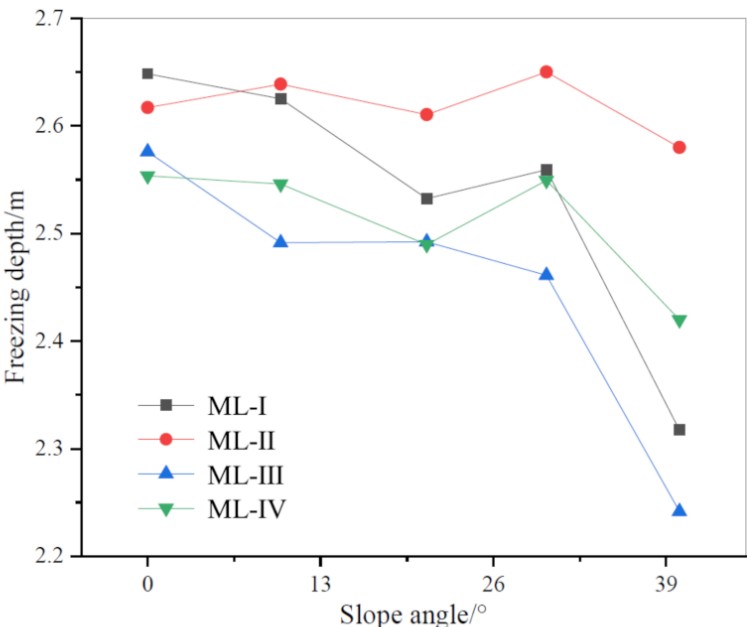

**Figure 12.** The variation law of freezing depth with slope angle.

3.2.3. Influence of Overburden Thickness on Temperature Field

Figures 13 and 14 show the nephogram of the temperature field of surrounding rock under the influence of overburden thickness. It can be seen whether during the melting or freezing period, when the overburden thickness is less than 15 m, there is a connection area between the surface temperature and the temperature of surrounding rock near the tunnel wall, located on the left side of the tunnel arch. For the 5 m thickness, the temperatures in the connection area are 20~21 °C and −8~−9 °C for the melting and freezing period, respectively, while for 11.6 m thickness, the temperatures are 7~8 °C and 2~3 °C, respectively. When the overburden thickness is greater than 15 m, the surrounding rock temperature field near the ground surface and the tunnel surrounding rock temperature field become independent of each other. There is a "isolated temperature zone" between the two temperature fields, with a temperature of 6–7 °C. The temperature of "isolated temperature zone" gradually evolves toward the surface and the tunnel wall, and eventually connects with the temperature fields on both sides.

The distribution law of tunnel temperature field on the monitoring line has also been studied under the influence of different overburden thicknesses. It can be observed that the temperature of the surrounding rock decreases with the distance from the tunnel wall in the melting period, and shows an inverse trend in the freezing period (Figure 15). Compared to the deep buried side, the temperature of the surrounding rock on the shallow buried side is greatly affected by the air temperature, especially with the overburden thickness of 5 m, the temperature evolution trend of the tunnel surrounding rock on shallow buried side is significantly different from other conditions. In the melting period, the temperature curves corresponding to 5 m thickness on monitoring line I and III tend to stabilize at 6 m and 7 m, respectively, followed by a rebound phenomenon. In the freezing period, the temperature curves corresponding to 5 m thickness on monitoring line I and III tend to stabilize at 5 m and 6 m, respectively, followed by a downward trend. These phenomena indicate that when the thickness of the covering layer is small, the air temperature has a significant effect on the temperature of the rock surrounding the tunnel, and there is an interaction effect between the surface air temperature and the internal air temperature of the tunnel. For the deeply buried side, the temperature of the rock surrounding the tunnel is essentially unaffected by the surface air temperature, and its distribution is dominated by the internal air temperature of the tunnel.

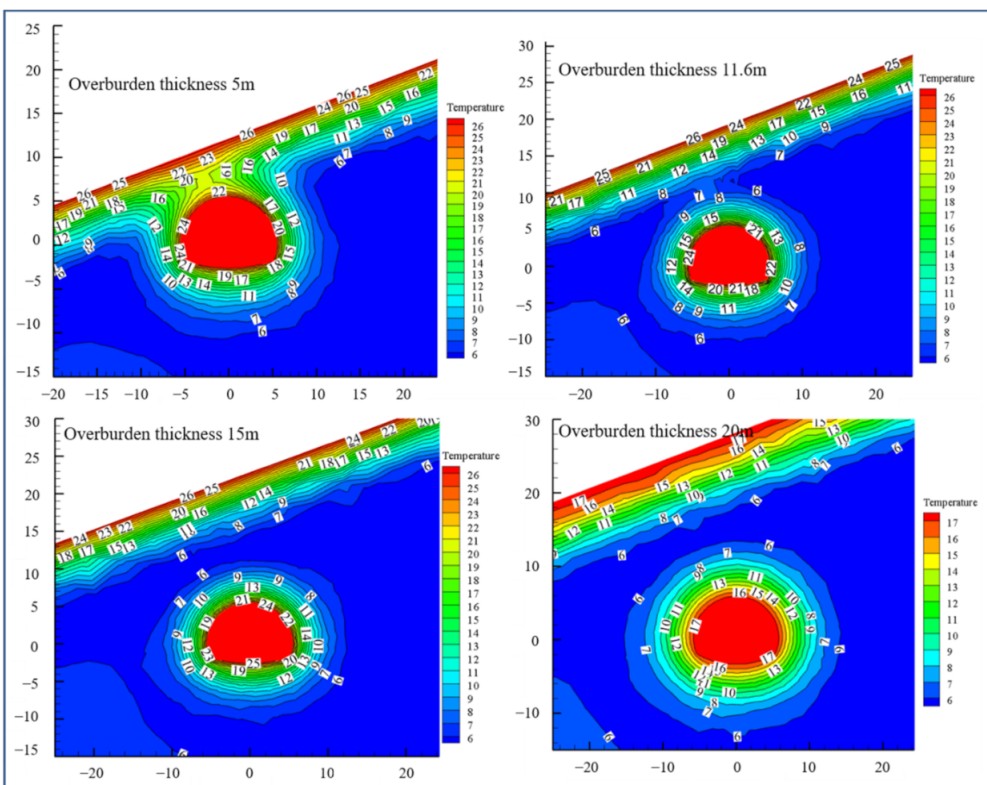

**Figure 13.** Temperature field of surrounding rock with different overburden thickness in melting period.

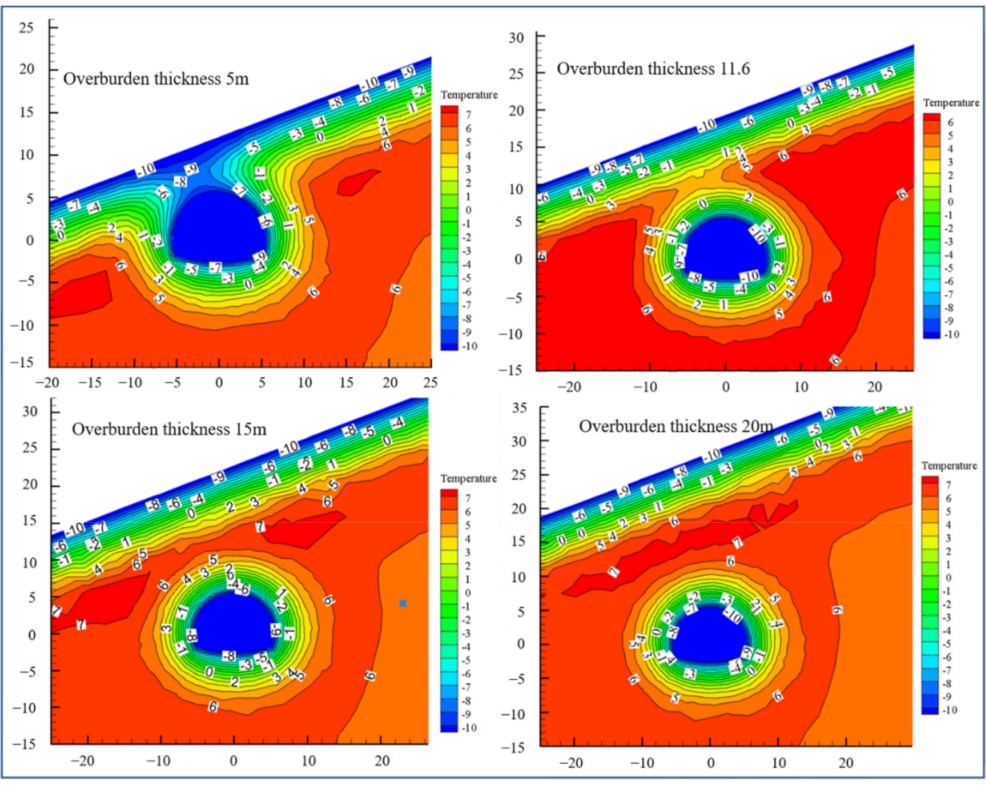

**Figure 14.** Temperature field of surrounding rock with different overburden thickness in freezing period.

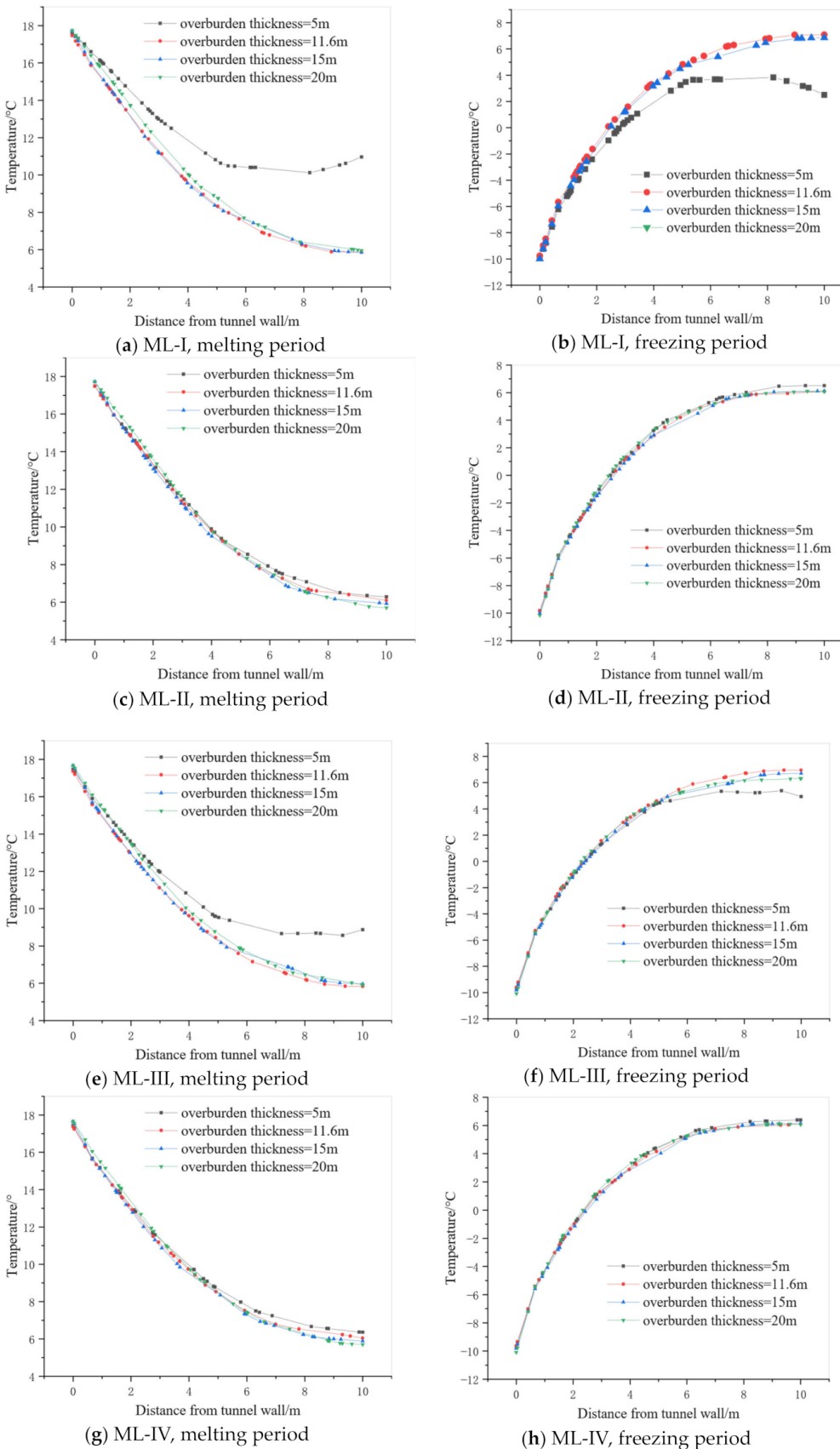

**Figure 15.** Temperature of different monitoring lines in melting and freezing period.

In summary, the temperature of the surrounding rock on the deep buried side is basically not affected by changes in the cover layer, while it on the shallow buried side is affected by changes in the cover layer thickness due to its proximity to the surface, and its impact depends on the distance from the surface. According to the temperature distribution law of freezing period, the change law of surrounding rock freezing depth under the influence of different overburden thickness is determined based on 0 °C. As shown in Figure 16, the freezing depth of the surrounding rock in the tunnel remains almost constant for different overburden thicknesses, and only slightly increases for a thickness of 5 m, probably due to the effect of the surface air temperature.

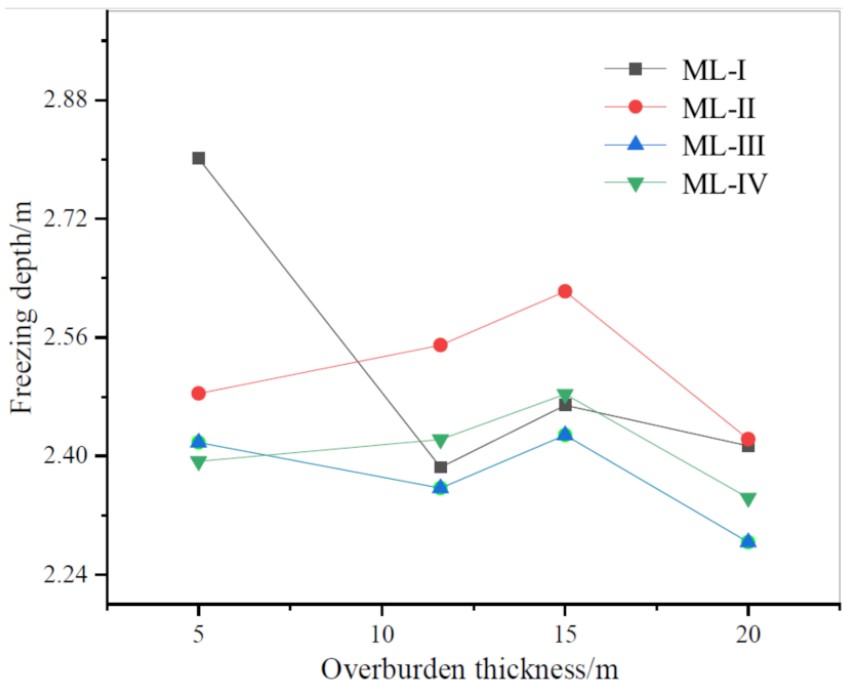

**Figure 16.** The variation law of freezing depth with overburden thickness.

The temperature field of surrounding rock of tunnel is the core issue for tunnel construction in cold regions. Especially in topographic bias tunnel, the temperature distribution of surrounding rock is extremely complex, which is affected by various factors such as terrain, burial depth, climate change [28,29]. This study found that the terrain slope angle significantly affect the distribution pattern of the surrounding rock temperature field, causing the temperature field to deflect toward the shallow buried side and changing the freezing depth of the surrounding rock. However, the effect of tunnel buried depth on the temperature field of the surrounding rock is relatively small, especially when the burial depth is larger than 15 m, where the surface air temperature essentially does not affect the temperature field of the surrounding rock in the tunnel. Therefore, in practical engineering construction, it is important to pay attention to the distribution law of the temperature field in a shallow buried terrain-biased tunnel.

## 4. Conclusions

This paper focuses on temperature field of surrounding rock of topographic bias tunnel in seasonally frozen regions. Taking the HTG tunnel as an example and using integrated method combining on-site monitoring and numerical analysis methods, the spatiotemporal evolution law of the surrounding rock temperature field of the HTG tunnel under actual working conditions is studied, and the influence mechanism of different terrain factors (slope angle and overburden thickness) on the surrounding rock temperature field is explored. The main conclusions are summarized as follows:

1.  Based on the section RK315+710 of the HTG tunnel, a three-dimensional numerical model was established. After comparison, the numerical simulation results were in good agreement with the monitoring results, verifying the feasibility of the turbulence theory, convective heat transfer, and wall function method.

2.  The increase in terrain slope angle can lead to a transformation of the connection area between the tunnel surrounding rock temperature and the surface temperature, and the connection area will deflect toward the shallow buried side (arch shoulder). However, an increase in the terrain slope angle does not lead to an increase in the freezing depth, which could be the result of a combination of the surface air temperature and the internal air temperature of the tunnel.

3.  As the overburden thickness gradually increases from 5 m to 20 m, the surface temperature and the tunnel surrounding rock temperature gradually change from interconnected to separate. When the overburden thickness exceeds 15 m, the temperature field of the tunnel surrounding rock is basically not affected by the surface air temperature, and there is an "isolated temperature zone" between the temperature field of the surrounding rock and the surface, with a temperature of about 6~7 °C. The freezing depth of the surrounding rock remains essentially constant under the influence of different covering thicknesses, increasing only slightly at a thickness of 5 m.

**Supplementary Materials:** The following supporting information can be downloaded at: https://www.mdpi.com/article/10.3390/w15112060/s1, Table S1: Numerical simulation results of surrounding rock temperature.

**Author Contributions:** Conceptualization, T.Z.; Methodology, T.Z.; Investigation, X.X.; Resources, X.X.; Writing—original draft, W.T.; Writing—review & editing, W.T.; Supervision, J.L.; Funding acquisition, H.W. All authors have read and agreed to the published version of the manuscript.

**Funding:** This work was supported by Guizhou Provincial Science and Technology Projects (No. QKHJC-ZK[2022]YB104).

**Data Availability Statement:** The data presented in this study are available on request from the corresponding author.

**Acknowledgments:** The authors thank the reviewers for their valuable suggestions.

**Conflicts of Interest:** The authors declared that they have no conflicts of interest.

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
