# Peer review of "Distribution Pattern and Influencing Factors for the Temperature Field of a Topographic Bias Tunnel in Seasonally Frozen Regions"

_water, doi:10.3390/w15112060_

Round 1
Reviewer 1 Report
The authors studied numerically turbulent fluid flow and heat transfer in the shallow buried bias section of the HTG tunnel using commercial COMSOL Multiphysics software. The standard k-eps model was used for analysis. Effects of governing parameters were studied. This paper can be published after major revision with following comments.
1. Numerical analysis part should be improved with partial differential equations for velocity and temperature.
2. Initial and boundary conditions should be formulated mathematically.
3. What is the reason for the standard k-eps model? Did you analyze other turbulence models?
4. Mesh sensitivity analysis is necessary. What time step and coordinates steps were used?
5. Description of the used difference schemes should be included.
6. Validation part is necessary. The authors should compare obtained results with experimental data.
7. There are some typos within the text.
7. There are some typos within the text.
Author Response
- Numerical analysis part should be improved with partial differential equations for velocity and temperature.
Response: The partial differential equations for velocity and temperature has been added in the Lines 108-129.
- Initial and boundary conditions should be formulated mathematically.
Response: We have added the relevant expressions in lines 191-195.
- What is the reason for the standard k-eps model? Did you analyze other turbulence models?
Response: We have investigated several models of air turbulence in our preliminary work, such as the algebraic yPlus model, the Spalart-Allmaras model, the k-ε two-equation model, the k-ω two-equation model, the SST turbulence model and the v2-f turbulence model. We compared and analyzed the advantages and disadvantages of these methods and their suitability for this study, and excluded the algebraic yPlus model, the Spalart-Allmaras model and the v2-f turbulence model. Based on the aforementioned boundary conditions of the paper, we set a constant type of inlet boundary and pressure boundary, therefore, the k-ε equation model is more advantageous in terms of computational speed and memory consumption, so we finally chose the k-ε two-equation model.
- Mesh sensitivity analysis is necessary. What time step and coordinates steps were used?
Response: We similarly recognized the importance of grid size in the presence of fluid. As mentioned in the references [27], and recommended by software engineers, we have chosen a grid size of 0.02m~0.1m. as shown in the Figure 1, for the air flow tunnel boundary, and a larger grid for the rock mass in this paper, which is 0.4m~5m.
Figure 1 Partial grid
The unit of t in the temperature function in this paper is days, but the default unit of time in the simulation software is seconds, so we have used a time step of 86400 seconds, which is one day. The above content has also been added to lines 201-208 of the revised manuscript.
- Description of the used difference schemes should be included.
Response: If it is difference schemes for different topographical factors, we have made a relevant description in the manuscript in lines 203-204.
- Validation part is necessary. The authors should compare obtained results with experimental data.
Response: Your suggestion is very reasonable. We have compared the monitoring results with the numerical analysis results in lines 254-260 of the revised manuscript, verifying the accuracy of the numerical model.
- There are some typos within the text.
Response: Thank you for your valuable and thoughtful comments. We have carefully checked and improved the English writing in the revised manuscript.

Reviewer 2 Report
Report: Water-2389224
Title: Distribution pattern and influencing factors for the temperature field of a topographic bias tunnel in seasonally frozen regions
This article is the study of highway tunnels through distribution and influencing factors for the temperature field. Numerical results are presented based on site monitoring data. Overall, the article presents publishable results, but a minor revision is needed. Detailed comments are provided below that need to be addressed.
1. Add the novelty of the problem at the end of the introduction.
2. Add recent papers related to the study.
3. Add source/reference to Table 1.
4. Add Tables to show the obtained data.
5. Can the authors explain, why they considered very old data (2015-2016), why not recent one?
Conclusion should be written in a proper way. Start with your study and then summarized.
English is fine only need moderate changes.
Author Response
- Add the novelty of the problem at the end of the introduction.
Response: Your suggestion is very reasonable, and we have added the innovative points of this article at the end of introduction (Line 77-81).
- Add recent papers related to the study.
Response: Thank you for your suggestion. We have supplemented multiple recent literature related to this study.
- Add source/reference to Table 1.
Response: Thank you for your suggestion. We have added reference to Table 1.
- Add Tables to show the obtained data.
Response: Thank you very much for your suggestion. Regarding the "obtained data", if you refer to the experimental results of numerical analysis, it has be transmitted to the system as a supplementary document.
- Can the authors explain, why they considered very old data (2015-2016), why not recent one?
Response: The HTG tunnel (RK315+710) was first built in 2015-5-8, and this monitoring work mainly focuses on the spatiotemporal evolution of the surrounding rock temperature field during the construction period and early operation of the tunnel, in order to guide the safe construction and operation of the project. In the later stage, due to sensor failure and other reasons, the monitoring data was missing or the accuracy was insufficient, so the tunnel was no longer monitored. All the data used in this article are the monitoring results from 2015 to 2016. Due to the main focus of this article on exploring the spatiotemporal evolution and influencing factors of temperature field in terrain biased tunnels, the monitoring time will not affect the core theme of this article.
- Conclusion should be written in a proper way. Start with your study and then summarized.
Response: Your suggestion is very reasonable, and we have made improvements to the conclusion of this article.
